# Role and effects of zinc supplementation in HIV-infected patients with immunovirological discordance: A randomized, double blind, case control study

**Macarena Silva**[1,2,3]*, **Carmen G. Montes**[4], **Andrea Canals**[5,6], **Maria J. Mackenna**[4,5,6,7], **Marcelo Wolff**[1,2,3]

**1** Arriarán Foundation, Santiago, Chile, **2** Department of Infectious Diseases, San Borja-Arriarán Hospital, Santiago, Chile, **3** School of Medicine, University of Chile, Santiago, Chile, **4** Nutrition and Food, Mention in Human Nutrition, Institute of Nutrition and Food Technology, University of Chile, Santiago, Chile, **5** Biostatistics, Public Health School, University of Chile, Santiago, Chile, **6** Academic Direction, Santa Maria Clinic, Santiago, Chile, **7** Nutrition Department, Las Condes Clinic, Santiago, Chile

* msilva@fundacionarriaran.cl

**Data Availability Statement:** All relevant data are within the manuscript and its Supporting Information files.

## Abstract

### Introduction

It has been estimated that between 15% and 18% of patients who start antiretroviral therapy (ART) do not achieve a successful immune recovery despite complete virological suppression. In the literature this phenomenom is known as poor immune recovery or immunovirological discordance (IVD). Zinc has an immunomodulatory role associated with T lymphocytes and its supplementation could enhance immune recovery.

### Objective

To determine if zinc supplementation on IVD patients prevents immune failure after 12 months of supplementation. Secondary objectives were to determine serum zinc levels in HIV patients with and without IVD and the frequency of hypozincemia in discordant patients.

### Method

We reviewed the historical record of patients under care at Arriarán Foundation. Following inclusion criteria were defined: 1) age $\geq$ 18 years, 2) standard ART (three effective drugs) for at least 18 months, 3) virologically suppressed for 12 months, 3) persistence of CD4 count $\leq$200 cells/mm$^3$ and/or increase $\leq$ 80 cells/mm$^3$ after one year of viral undetectability. A control group was assigned paired 1:1 by sex, age (± 2 years) that did achieved an increase of CD4> 350 cells/ mm$^3$. In both groups plasma zinc levels were determined. In a later phase, patients with IVD were randomized to receive zinc (15 mg daily) versus placebo. Patients were followed for 12 months with CD4 count, viral load and zinc levels determinations every 4–6 months.

**Funding:** SAVAL laboratory participated only in the preparation of tablets.

**Competing interests:** The authors have declared that no competing interests exist.

## Results

A total of 80 patients, 40 patients with IVD criteria and 40 controls were included. 92.5% were men, and age average was 47.5 years. The median baseline CD4 was 189 cells/mm$^3$ (71–258) in the cases vs. 552.5 cells/ mm$^3$ (317–400) in the control group with a median increase at the end of the study of 39 cell/mm$^3$ and 19 cell/mm$^3$ respectively. There was no difference in baseline plasma zinc levels between both groups (81.7 + 18.1 in cases versus 86.2 + 11.0 in controls). In the 40 patients with IVD, the median absolute increase in CD4 after annual zinc supplementation was 31.5 cells/mm$^3$ in the treated group versus 50 cells/ mm$^3$ in the placebo group, this difference being statistically not significant (p = 0.382).

## Conclusions

Patients with IVD have plasma zinc levels similar to those who achieve adequate immune recovery. Zinc supplementation in IVD patients showed a statistically non-significant difference in in CD4 levels between cases and controls. The results warrant a comparative study with a larger number of patients.

## Introduction

Antiretroviral therapy (ART) has significantly improved the morbidity and mortality of HIV-infected patients, increasing life expectancy to approximately 90% after 5 years, depending on the baseline immunosuppression at which ART is started [1]. Concomitantly, the proportion of patients who start treatment at advanced stages decreased by half from 2002 to 2010 [2–5].

The proportion of patients that start ART and do not achieve a successful immune recovery, despite complete virological suppression, fluctuates between 15 and 18%. In the literature, this phenomenon is known as poor immune recovery, immunologic failure, or immunovirological discordance (IVD) [6–12]. Its prevalence varies according to the definition, from 8%, using less than 200 cells/mm$^3$ CD4 + T lymphocytes (CD4) after 6 years of ART as a criterion [6], to 25%, based on an increase of less than 50 cells/mm$^3$ after 6 months of ART [7, 8]. Recent data reported a prevalence of 15% based on the persistence of CD4 at less than 200 cell/mm$^3$ after 3 years of ART [9].

Among the predisposing factors for poor immune recovery, the level of baseline immuno-suppression [9] and age stand out. As a consequence, patients with IVD have been shown to have a higher overall mortality [9–11] in both defining and non-defining AIDS events, especially hepatitis and cancer not associated with HIV [9]. Zinc is known to have an important immunomodulatory role, as a determinant of immune function [12]. Among the general population, low zinc levels fluctuate between 20 and 40%, resulting in a decrease in the proliferation of T lymphocytes, humoral and cellular immunity, lymphopenia, and thymic atrophy [13–15]. Supplementation has been successfully used for immune failure prevention in HIV patients, reducing up to 4 times the risk of IVD [14, 16, 17].

In Chile, the magnitude of zinc deficiency among HIV-infected patients is unknown, with no studies evidencing whether patients on ART and IVD could have lower serum zinc levels than patients without IVD. The main purpose of this study was to determine if zinc supplementation on IVD patients prevents immune failure after 12 months of supplementation. Secondary objectives were to determine serum zinc levels in HIV patients with and without IVD and the frequency of hypozincemia in discordant patients.

## Method

This study was approved by University of Chile School of Medicine. Research Ethics Committee on Human Being, Written consent. Approval certificate N° 1092011"

IVD was defined as the lack of an increase in CD4 cells count over 200 cells/mm$^3$ after 12 months of successful ART, indicating an undetectable viral load during their first year of treatment.

This study had two stages. In the first stage, plasma zinc levels of IVD patients were compared with control subjects who achieved immune recovery, this meant an increment >350 of CD4 cells/mm$^3$ after 12 months of successful ART. Case and controls were paired by age and sex in a 1:1 ratio.

In a second stage, we randomized IVD patients to receive zinc sulfate at a dose of 15 mg/day versus placebo for a total follow-up of 12 months.

For this, we reviewed the historical record of patient under care at Arriarán Foundation, the largest HIV care center in Chile. The following inclusion criteria were defined: 1) confirmed HIV infection; 2) >18 years; 3) on standard ART (three effective drugs) for at least 18 months; 4) viral undetectability (< 50 copies/ml or isolated transient increases in viremia of less than 1000 copies/m)l; and 5) persistence of CD4 <200 cells/mm$^3$ at enrollment or an increase of less than 80 cells/mm$^3$ after one year of viral undetectability, in which the latter case maintained an CD4 <250 cells/mm$^3$ at enrollment.

Patients with the following conditions were excluded from the study: 1) pregnancy or lactation; 2) active or recent opportunistic infection (3 months); 3) neoplastic disease with or without chemotherapy/ radiotherapy; 4) zinc intake as a supplement during the last 6 months (multivitamins were not excluded due to the low zinc content); 5) current or recent use (<3 months) of other myelosuppressive drugs; or 6) other possible causes of immunosuppression, such as diabetes mellitus, chronic liver damage, kidney failure stage V chronic, or chronic steroid therapy (>15 mg/day of prednisone for more than one month).

Each of the patients signed an informed consent, approved by the Research Ethics Committee on Human Being, University of Chile School of Medicine. Demographic data, coinfection with Hepatitis B or C, current antiretroviral therapy, smoking, and anthropometric (weight, height, BMI) characteristics were recorded at the baseline visit.

In addition, a 24-hour reminder food survey was conducted to calculate zinc intake. Plasma zinc levels were performed using the atomic absorption spectrophotometry technique and processed in the micronutrient laboratory of the Institute of Nutrition and Food Technology (INTA). Hypozincemia was defined according to the criteria of the International Zinc Nutrition Consultative Group (IZING), which considers plasma zinc cut-off levels at 74 μg/dl for men and 70 μg/dl for women.

Patients were monitored every 4 months, with CD4 cell measurement, viral load, complete blood count, biochemical profile, and plasma zinc levels recorded. At each visit, four vials were administered, each prepared with the number of tablets for one month and the medication count of the vials delivered by the patient was recorded. A SMAQ survey was conducted to determine adherence.

We calculated the sample size needed to assess whether the relative risk of a CD4 increase above 30%. We based on the percentages of CD4 increase greater than 30% from de study of Asdamongkol et al (46.1% in patients with Zinc and 23.53% in patients with placebo), with a power of 80% and a confidence level of 95%. According to this calculation, a sample size of 28 patients per group was required. The calculation was made with nMaster 2.0 software.

Power was calculated for the comparison of the increasing average CD4 between cases (zinc) and controls (placebo), obtaining a value of 100%. This result indicates that, although

we did not reach the previously calculated sample size (28 patients per group), the sample size obtained was sufficient to assess whether the increase in CD4 levels is different between cases and controls.

Descriptive analyses on the variables of interest were performed. For qualitative variables, percentage frequencies of the different categories of the variable were calculated and for quantitative variables, average, standard deviation, median, and range were calculated. The Shapiro-Wilk Test was used to study the normality of quantitative variables. To compare the distribution of a quantitative variable in treatments and controls, T Test was used in the case of normal variables or the Wilcoxon-Mann-Whitney´s in the case of non-normal variables. To analyze the association between categorical variables, Fisher's Exact Test was used The data were analyzed with the STATA 14 statistical software and a statistical significance of $\alpha = 0.05$ was considered an intention to treat (ITT) mode study.

DOI link: https://dx.doi.org/10.17504/protocols.io.bpqimmue

## Results

A total of 80 patients were included; 40 patients met immunovirological discordance criteria and 40 controls were matched 1:1 by sex and age. The baseline characteristics of cases (patients with IVD) and controls (patients without IVD) are described in Table 1.

The percentage of men was 92.5% and the average age was 47.5 years old. Nutritional status, evaluated as BMI, was similar in both groups, at 24 (22.1–26.6) in treatments and 24.2 (22.7–26.6) in controls. Zinc intake was also similar, at 7.7 (6.4–13.1) and 10 (7.6–12.1) mg/day, respectively. The percentage of patients with HBV was 15.4% in treatments and 7.9% in

**Table 1. Baseline characteristics of subjects with IVD and controls.**

| Characteristic[*] | IVD (n = 40) | No IVD (n = 40) | P–value [a] |
|---|---|---|---|
| Male sex (%) | 37 (92.5) | 37 (92.5) | 1.000 |
| Age–years | 47.9 ± 9.8 | 47.2 ± 10.2 | 0.764 |
| Time since HIV diagnosis–months | 72.8 (37.6–117.7) | 132.1 (91.8–174.2) | <0.001 |
| Time on ART–months | 45.1 (25.6–96.1) | 120.6 (69.3–143.7) | <0.001 |
| Hepatitis B | 6 (15.4) | 2 (7.9) | 0.481 |
| Hepatitis C | 1 (2.9) | 1 (2.8) | 0.739 |
| ART (%) | | | |
| NN+N | 22 (55) | 28 (70) | 0.248 |
| NN+PI | 11 (27.5) | 8 (20) | 0.600 |
| NN+INSTI | 7 (17.5) | 4 (10) | 0.518 |
| Viral load <50 (copies/ml) | 40 (100%) | 40 (100%) | - |
| Baseline CD4 (cells/mm$^3$) | 189 (150–208.5) | 552.5 (463.5–690.0) | <0.001 |
| Final CD4 | 223 (197–256) | 582 (506–720) | <0.0001 |
| Δ CD4 (final CD4-baseline CD4) | 39 (12–80) | 19 (36–100) | 0.1433 |
| Zinc intake (mg/day) | 7.7 (6.4–13.1) | 10 (7.6–12.1) | 0.208 |
| BMI index (kg/m$^2$) | 24 (22.1–26.6) | 24.2 (22.7–26.6) | 0.503 |
| Zinc level (ug/dl) | 81.7 ± 18.1 | 86.2 ± 11.0 | 0.266 |
| Hypozinchemia (%) | 12 (32.4) | 2 (6.5) | 0.008 |

[*]Quantitative variables with normal distribution: mean ± s.d, quantitative variables without normal distribution: median(IQR), categorical variables: n (%)

[a] t test, Wilcoxon-Mann-Whitney´s test, Ficher's exact test

Abbreviations: IVD immunovirologic discordance; ART antiretroviral therapy; NN non-nucleoside reverse transcriptase inhibitor; N nucleoside reverse transcriptase inhibitors; PI protease inhibitor; INSTI integrase inhibitor; BMI body mass index

controls (p = 0.481) and with Hepatitis C, 2.9% and 2.8% respectively. The type of antiretroviral therapy indicated was also comparable in both groups. Patients with IVD had a shorter time from diagnosis, with 6.1 versus 11 years (p = 0.0003) and consequently, fewer months on ART [45.1 (25.6–96.1) vs 120.6 (69.3–143.7); p<0.001]. The number of CD4 T lymphocytes at the start of ART was significantly lower in patients with IVD [36 (19.0–110.5) cells/mm$^3$] than in non-discordant patients [187 (82–273) cells/mm$^3$]; (p = <0.0001). During the 12 months study period, the median of increment on CD4 cells between cases and controls was 39 cell/mm$^3$ and 19 cell/mm$^3$ with no differences between them (p 0.1433)

The average serum zinc level in the total population was 87.8 µg/dl. The percentage of patients with hypozincemia was significantly higher in the group of patients with IVD, with 12 ug/dl (32.4%) vs 2 ug/dl (6.5%), respectively (p = 0.008); with no differences in the average of basal zinc levels between both groups. (81.7±18.1 ug/dl in treatments and 86.2±11 µg/dl in controls (p = 0.266)

A sub-analysis of baseline characteristics was compared in IVD patients with hypozincemia and normozinchemia (Table 2).

There were no significant differences in terms of age, months on ART, prevalence of hepatitis B or C, type of ART, level of immunosuppression at baseline, body mass index, or zinc intake.

Of the 40 IVD patients, 19 were randomized to receive zinc supplementation and 21 to placebo. All baseline characteristics were comparable.

After 12 months of supplementation, CD4 values at the end of the period were 223 (197–256) cells/mm$^3$ for all patients with IVD, with a median annual increase of 39 cells/mm$^3$ (12–80 cells/mm$^3$). At the end of the study, patients in the zinc group reached CD4 values of 222.5 cells/mm$^3$ (202–260), versus 223 cells/mm$^3$ (197–251) in the placebo group. No difference was found in the CD4 increases between both groups [zinc 31.5 cells/mm$^3$ (20–80) and placebo 50 cells/mm$^3$ (22–69; p = 0.382). The median plasma zinc level after supplementation in the

**Table 2. Baseline characteristics of IVD subjects according to zinc level.**

| Characteristic* | Hypozinchemia (n = 12) | Normozinchemia (n = 25) | P-value[a] |
|---|---|---|---|
| Male sex–% | 11 (91.7) | 23 (92) | 0.704 |
| Age–years | 44.5 (40.5–48) | 51 (44–55) | 0.060 |
| Time since HIV diagnosis–months | 63.8 (45.7–88.6) | 68.9 (32.7–120.5) | 0.559 |
| Time on ART–months | 33 (24.8–51.9) | 47 (32.7–107.5) | 0.116 |
| ART (%) | | | |
| NN+N | 5 (41.7) | 15 (60) | 0.482 |
| NN+PI | 5 (41.7) | 6 (24) | 0.443 |
| NN+INSTI | 2 (16.7) | 4 (16) | 1.000 |
| Hepatitis B (%) | 2 (16.7) | 3 (12.5) | 0.549 |
| Hepatitis C (%) | 1 (10) | 0 (0) | 0.323 |
| Baseline CD4 (cells/mm$^3$) | 179.5 (146–197.5) | 192 (156.0–216.0) | 0.455 |
| BMI (kg/m$^2$) | 24.0 (20.4–25.6) | 24.9 (22.4–27.1) | 0.638 |
| Zinc intake (mg/d) | 8.2 (6.4–13.8) | 7.8 (6.0–13.1) | 0.888 |
| Zinc level (ug/dl) | 65 (56.3–70) | 92.5 (80.0–100) | <0.001 |

*Quantitative variables: mean ± s.d., categorical variables: n (%)

[a] Wilcoxon-Mann-Whitney's test, Fisher´s exact test

Abbreviations: IVD immunovirologic discordance; ART antiretroviral therapy; NN non-nucleoside reverse transcriptase inhibitor; N nucleoside reverse transcriptase inhibitors; PI protease inhibitor; INSTI integrase inhibitor; BMI body mass index

**Table 3. Characteristics of IVD patients randomized to zinc or placebo.**

| Characteristic * | Zinc (n = 19) | Placebo (n = 21) | P-value[a] |
|---|---|---|---|
| Male–(%) | 16 (84.2) | 21 (100) | 0.098 |
| Age–years | 46.0 (40.0–54.0) | 48 (44–55.0) | 0.489 |
| Time on ART–months | 40.2 (24.0–62.7) | 54.7 (32.7–107.5) | 0.189 |
| Hepatitis B (%) | 3 (15.8) | 3 (15.0) | 0.644 |
| Hepatitis C (%) | 0 (0) | 1 (5.6) | 0.529 |
| BMI (kg/m$^2$) | 24.0 (22.4–27.6) | 24.0 (21.5–26.4) | 0.591 |
| ART | | | |
| N+NN | 10 (52.6) | 12 (57.1) | 1.000 |
| N+PI | 6 (31.6) | 5 (23.8) | 0.727 |
| N+INSTI | 3 (15.8) | 4 (19.1) | 1.000 |
| Baseline CD4 (cells/mm$^3$) | 197 (177–216) | 180 (131–198) | 0.110 |
| Final CD4 (cells/mm$^3$) | 222.5 (202–260) | 223 (197–251) | 0.602 |
| Δ CD4 (final-baseline) Cells/mm$^3$ | 31.5 (2.0–80.0) | 50 (22–69) | 0.382 |
| Zinc intake (mg/d) | 8.2 (6.4–13.3) | 7.5 (6.4–12.8) | 0.970 |
| Hypozinchemia (%) | 6 (31.6) | 6 (33.3) | 0.593 |
| Baseline Zinc (ug/dl) | 85 (70–95) | 77.5 (70–95) | 0.670 |
| Final Zinc (ug/dl) | 105 (75–132.5) | 82.5 (68–87.5) | 0.285 |

*Quantitaive variables: median (IQR), categorical variables: n (%)

[a] Wilcoxon-Mann-Whitney´s test Fisher´s exact test.

Abbreviations: IVD immunovirologic discordance; ART antiretroviral therapy; NN non-nucleoside reverse transcriptase inhibitor; N nucleoside reverse transcriptase inhibitors; PI protease inhibitor; INSTI integrase inhibitor; BMI body mass index

group with zinc and placebo was 105 (75–132.5) and 82.5 (68.8–87.5) respectively (p = 0.285) (Table 3).

Within the group of patients assigned to receive zinc, immune recovery of patients with hypozincemia was analyzed and compared to those with normozinchemia; no statistical difference was identified (56 (11–59) and 31.5 (10–93.5); p = 0.752) (Table 4).

The supplemented hypozinchemic patients achieved normozinchemia by the end of treatment. During the follow-up, only one patient presented a severe adverse event due to gastrointestinal intolerance, which required suspension of treatment and who belonged to the zinc group. There was one ART failure and 3 dropouts during follow up period.

## Discussion

Zinc is a divalent cation of mainly catalytic action, obtained through the intake of marine and animal food sources [18], and that participates in a wide range of metabolic processes. The vast majority of zinc is found in musculoskeletal tissue. As absorption occurs in the small intestine,

**Table 4. Response of CD4 of supplemented patients accord to zinc level.**

| CD4$^+$ (cells/mm$^3$) * | Hypozinchemia (n = 6) | Normozinchemia (n = 13) | P- value |
|---|---|---|---|
| Baseline | 188.5 (179–224) | 193 (163.5–213) | 0.543 |
| Final | 235 (191–256) | 222.5 (206.5–261) | 0.752 |
| Δ CD4$^+$ | 56 (11–59) | 31.5 (10–93.5) | 0.752 |

*mean ± s.d. Abbreviations: Δ final CD4- Baseline CD4

its losses are on the gastrointestinal level through pancreatic, biliary, and intestinal secretions. Zinc deficiency originates as a result of inadequate dietary intake, especially in periods that increase the body's requirements, such as rapid growth, malabsorption, increased losses, and/or impediments to use zinc [19]. This can affect up to one-third of the general population, especially in developing countries and affects different populations according to lifestyle, age, and chronic diseases. Its antiviral role has been determined at various stages of the viral replication cycle. In vitro studies have shown free virus inactivation, inhibition of denudation, viral genome transcription, and interference in polyprotein processing. Its direct immunological function is not specified, but its status in the body may influence antiviral immunity [20]. The benefit of its supplementation has been mainly associated with a decrease in episodes of diarrhea and respiratory infections in children. A study in Tanzanian infants showed significant reduction in all causes of diarrhea and respiratory infections compared to multivitamins and a placebo [21]. Zinc has also been found to reduce inflammation biomarker levels (ej. sCD14, sTNF-RI) [22].

The concept of immune failure, immunovirological discordance, or poor immune recovery has been used interchangeably in the literature to define those patients who do not achieve adequate immune recovery after a certain period under successful ART. This has been defined according to the DHHS Panel on Antiretroviral Guidelines as an annual increase in CD4 of 50–150 cell/year. The lack of uniformity in its definition criteria has probably led to the dismissal of its frequency and thus, to possible therapeutic strategies.

In a systematic review on IVD that included more than 20 studies, a 2 to 3-fold increase in mortality was observed, considering high heterogeneity of its definition [23]. It has also been reported that the risk of developing AIDS-defining and non-defining events is 1.4 times higher, with an RR of 1.43 for presenting any of these events [24, 25]. In the study published by Hadadi on HIV patients with a CD4 count >200 cells/mm$^3$, a zinc deficiency of 44% was found, and after its supplementation, a reduction in opportunistic infections or an increase in the CD4 count was not demonstrated [26]. Conversely, Asdamongkol reported in a limited series, positive effects of zinc supplementation in IVD patients with hypozinchemia [27].

Our study is the first one to be conducted nationwide to determine zinc levels in HIV patients with and without immunovirological discordance and the first to associate hypozinchemia with this last condition. In this study, it was not possible to determine a benefit in immune recovery after zinc supplementation, both in normal and hypozinchemic patients. The study population was preferably male, with a BMI in the normal range (BMI 24), and adequate zinc intake based on nutritional recommendations for the general population (RIC between 7 and 13 mg per day). Patients with IVD were not in unfavorable nutritional conditions based on the measurement of BMI and their zinc plasma levels were similar to patients with complete immune recovery. IVD patients had lower baseline CD4 counts, which is expected since the higher the basal immunosuppression, the slower the immune recovery. The average age of the patients with IVD was higher than the general chilean HIV population.

Notably, one-third of the patients with IVD had hypozinchemia on admission. The basal levels were categorically lower, with a median of 65 mg/dl, a value considerably lower than the general population. Hypozincemia was not attributable to differences of zinc intake, based on the applied survey, to co-infection with HBV or other baseline characteristics but ART time exposure was significantly lower than control group. It is unknow the effect that accumulative ART time exposure could have on zinc levels. However, immune recovery, independent of zinc supplementation, was similar in both normal and hypozinchemic patients. A recently published systematic review [28] evaluated the effect of vitamin D, selenium and zinc supplementation in HIV adults and pediatric patients. The analysis showed some evidence in reducing diarrhea morbidity and in restoring immune function. However, of all the studies

included, only Asdamongkol reported benefit in immune recovery after zinc supplementation in the context of a low number of patients with hypozinchemia. No benefit was reported in immunological recovery, in terms of CD4 increase, in the rest of the studies included in the analysis [16, 29–32].

High frequency of HBV infection was observed in the group with immunovirological discordance compared to the control group (15.4 vs. 7.9%, respectively), which was not correlated with plasma zinc levels; that is, HBV coinfection was associated with slow immune recovery but not with hypozinchemia.

The relationship between zinc and hepatitis was previously observed but associated with HCV infection, showing that zinc naturally inhibits the inflammatory and antiviral effects of interferon lambda 3 (IFN-λ3), a protein strongly associated with tissue damage in chronic liver disease [33]. Additionally, it has benefits in reducing the risk of progression in a hepato-cellular carcinoma [33]. Zinc supplementation was considered safe in this study, with only one case of severe digestive intolerance and its administration, even in normozincemic patients, did not cause significant plasma increases. This is evidenced by its well-known adequate homeostasis between its absorption and elimination. There was one ART failure and 3 patients were lost after the follow-up at 6 months. The limitations of this study lie mainly in the sample size, given by the low frequency of this condition, and that other causes of immunosuppression were excluded in this group. Multicenter studies with a larger number of patients are required to be able to verify if there is a real benefit in zinc supplementation, either in patients with IVD or only in those with hypozinchemia.

## Supporting information

**S1 Data.**
(DO)

## Author Contributions

**Conceptualization:** Macarena Silva, Carmen G. Montes, Maria J. Mackenna, Marcelo Wolff.

**Data curation:** Macarena Silva, Carmen G. Montes.

**Formal analysis:** Macarena Silva, Carmen G. Montes, Andrea Canals, Marcelo Wolff.

**Funding acquisition:** Carmen G. Montes, Maria J. Mackenna.

**Investigation:** Marcelo Wolff.

**Methodology:** Macarena Silva, Carmen G. Montes, Andrea Canals, Maria J. Mackenna, Marcelo Wolff.

**Project administration:** Macarena Silva.

**Validation:** Andrea Canals, Marcelo Wolff.

**Writing – original draft:** Macarena Silva.

**Writing – review & editing:** Marcelo Wolff.

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
