## [Decision Letter · Decision Letter 0]

16 Sep 2020

PONE-D-20-19537

Role and Effects of Zinc Supplementation in HIV-infected Patients with Immunovirological Discordance: A Randomized, Double Blind, Case Control Study

PLOS ONE

Dear Dr. Silva,

Thank you for submitting your manuscript to PLOS ONE. After careful consideration, we feel that it has merit but does not fully meet PLOS ONE’s publication criteria as it currently stands. Therefore, we invite you to submit a revised version of the manuscript that addresses the points raised during the review process.

We look forward to receiving your revised manuscript.

Kind regards,

Giuseppe Vittorio De Socio, MD, PhD

Academic Editor

PLOS ONE

Journal Requirements:

2. Please provide additional details regarding participant consent. In the ethics statement in the Methods and online submission information, please clarify whether consent was informed.

4. Please include a copy of Table 5 which you refer to in your text on page 20

Reviewers' comments:

Reviewer's Responses to Questions

**Comments to the Author**

1. Is the manuscript technically sound, and do the data support the conclusions?

Reviewer #1: Yes

Reviewer #2: Partly

2. Has the statistical analysis been performed appropriately and rigorously? 

Reviewer #1: Yes

Reviewer #2: No

3. Have the authors made all data underlying the findings in their manuscript fully available?

Reviewer #1: Yes

Reviewer #2: Yes

4. Is the manuscript presented in an intelligible fashion and written in standard English?

Reviewer #1: Yes

Reviewer #2: Yes

5. Review Comments to the Author

Reviewer #1: In this paper Silva et al. investigate the role of zinc plasmatic levels and zinc supplementation in patients failing satisfactory CD4 recovery on virally-suppressive ART (IVD). The authors find higher proportion of hypozincemic patients in the IVD vs control group. However, they failed to find a positive effect of zinc supplementation.

The paper is well-written and straight forward, addressing an unexplored issue.

I find that a major bias in the patients population consists in the significant difference in the time on ART between IVD and controls, with IVD being on ART for significantly lower time. Lower ART time would mean lower possibility to achieve full immune recovery. The authors should either enroll a 3rd cohort of IVD with equal time on ART, or at least acknowledge this bias.

In Table 1 the authors report data on hepatitis B co-infection? Can the authors provide also data on HCV co-infection, which has been described to associate to low immune recovery on cART.

Reviewer #2: This study determined whether zinc supplementation in virologically suppressed patients who do not achieve complete immune recovery, prevents immune failure after 12 months of supplementation. The study included 80 patients, 40 with incomplete immune recovery (IVD) and 40 controls with CD4 cell count >350 cells/mm3. The mean plasma zinc levels were not different between the groups as baseline and were within the normal ranges. While the CD4 count in the incomplete recovery group increased more than in the control group, the difference was not significantly different.

The study is well planned and described. The investigators, however, did not provide sample size calculation and as they note, there may not have been sufficient number of study participants to show significant difference between the groups. Moreover, the manuscript does not show the HIV viral load of the patients and it is not clear if the IVD patients were virally suppressed. This may play a role in the incomplete immune recovery.

The mean plasma zinc levels were within normal range at baseline, with a few participants having hypozincemia. This contrasts with previous studies showing a positive effect of zinc supplementation on CD4 cell count of patients with low plasma zinc levels at baseline. While there were participants with very low zinc levels in this study, there were too few of them to make a judgement about the effectiveness of zinc supplementation in patients with low levels of zinc.

Another problem is that it is not clear whether the data was analyzed with intent to treat analyses. It also does not appear that the analyses controlled for baseline CD4 cell count which was significantly different between the groups. There is no table showing differences between the treatment and the placebo groups at the end of the study among all participants that started the study, only those with incomplete immune recovery. The results for the 40 participants who did not have incomplete immune recovery are not shown after 12 months of supplementation, only their baseline characteristics. Thus, this manuscript is basically about 40 people with IVD who were either supplemented with zinc or received placebo.

There are minor errors in English language than need to be corrected.

6. PLOS authors have the option to publish the peer review history of their article (what does this mean?). If published, this will include your full peer review and any attached files.

Reviewer #1: **Yes: **Giulia Marchetti

Reviewer #2: No

---

## [Author Response · Author response to Decision Letter 0]

4 Dec 2020

We thank the reviewers and editorial comments for their critical assessment of our manuscript. In the following we address their concerns point by point. 

Reviewer 1

Point 1.1: “I find that a major bias in the patients population consists in the significant difference in the time on ART between IVD and controls, with IVD being on ART for significantly lower time. Lower ART time would mean lower possibility to achieve full immune recovery. The authors should either enroll a 3rd cohort of IVD with equal time on ART, or at least acknowledge this bias.”

R: We recognize de bias since the longer time of ART exposure, the greater immune recovery achieved, especially during the first 5 years. For this reason, we included a comparative analysis of immune recovery based on final CD4 cell count for both groups (IVD and controls). We found no differences on the median CD4 cell count at the end of the study (IVD: 39 cell/mm3, controls: 19 cell/mm3; p 0,1433). Furthermore, we do not know if hypozinchemia observed in the IVD group could be related to the time of ART exposure. This aspect was incorporated on the discussion.

Point 1.2: “Can the authors provide also data on HCV co-infection, which has been described to associate to low immune recovery on ART”.

R: We added the HCV frequency for both groups on table 1 (IVD: 2.9%; controls: 2.8%; p 0.739)

Reviewer 2

Point 2.1: “The investigators, however, did not provide sample size calculation and as they note, there may not have been sufficient number of study participants to show significant difference between the groups” 

R: We calculated the sample size needed to assess whether the relative risk of a CD4 cell count increase above 30%. We were based on the percentages of CD4 cell increase greater than 30% from de study of Asdamongkol et al (46.1% in patients with zinc and 23.53% in patients with placebo), with a power of 80% and a confidence level of 95%. According to this calculation, a sample size of 28 patients per group was required. The calculation was made with nMaster 2.0 software.

Power was calculated for the comparison of the increasing average CD4cell count between cases (zinc) and controls (placebo), obtaining a value of 100%. This result

indicates that, although we did not reach the previously calculated sample size (28

patients per group of zinc and placebo), the sample size obtained was sufficient to assess whether the increase in CD4 cell levels is different between cases and controls.

This item is added on Methods.

Point 2.2 : “the manuscript does not show the HIV viral load of the patients and it is not clear if the IVD patients were virally suppressed. This may play a role in the incomplete immune recovery”

R: All patients had to be virologically suppress to enter the study. This item is added on table 1 and reinforced on discussion. 

Point 2.3 “This contrasts with previous studies showing a positive effect of zinc supplementation on CD4 cell count of patients with low plasma zinc levels at baseline. While there were participants with very low zinc levels in this study, there were too few of them to make a judgement about the effectiveness of zinc supplementation in patients with low levels of zinc”

R: The study of Asdamongkol is the only one that has demonstrated a benefit of zinc supplementation on hypozinchemic patients and also included a low number of patients (hypozinchemia:12 (5 supplemented), normozinchemia:18 (8 supplemented) Their findings were very similar to ours (hypozinchemia:12 (6 supplemented); normozinchemia: 25 (13 supplemented) , but our p value was not satistically significant. The table in the article is above: 

(table can be seen on file attached "Response to Reviewers") 

Asdamongkol N, Phanachet P, Sungkanoparph S. Low plasma zinc levels and immunological responses to zinc supplementation in HIV‐in- fected patients with immunological discordance after antiretroviral ther- apy. J Int AIDS Soc. 2012;15 Suppl 4:152‐3. 

Our findings: (the proper format of the table can be seen on file attached "Response to Reviewers"

Table 4. Response of CD4 of supplemented patients accord to zinc level. 

(12 hypozinchemia: 6 supplemented; 25 normozinchemia 13 supplemented)

CD4+ (cells/mm3) * Hypozinchemia(n=6) Normozinchemia(n=13) P- value

Baseline 188.5 (179-224) 193 (163.5-213) 0.543

Final

235 (191-256)

222.5 (206.5-261) 0.752

CD4+ 56 (11-59) 31.5 (10-93.5) 0.752

Futhermore, on a recent review neither Baum, Bobat, Green, Range or Reich found differences on increase of CD4 cell count after zinc supplementation at any zinc blood level. (AIDS Rev. 2020;22:1-10) This aspect is added on the discussion and reference at bibliography. 

Point 2.4: “another problem is that it is not clear whether the data was analyzed with intent to treat analyses”

R: the data was analyzed with intent to treat. This item is reinforced on the Methods. 

Point 2.5” It also does not appear that the analyses controlled for baseline CD4 cell count which was significantly different between the groups. There is no table showing differences between the treatment and the placebo groups at the end of the study among all participants that started the study, only those with incomplete immune recovery”

R: we added on table 1 the immune recovery at the end of the study of IVD and control patients showing no differences. This aspect was reinforced on the discussion.

Thank you for your consideration of this manuscript.

Kind regards,

Macarena Silva, MD 

Infectious Diseases 

Hospital San Borja Arriarán 

University of Chile School of Medicine 

Santiago, Chile

---

## [Decision Letter · Decision Letter 1]

17 Dec 2020

Role and Effects of Zinc Supplementation in HIV-infected Patients with Immunovirological Discordance: A Randomized, Double Blind, Case Control Study

PONE-D-20-19537R1

Dear Dr. Silva,

We’re pleased to inform you that your manuscript has been judged scientifically suitable for publication and will be formally accepted for publication once it meets all outstanding technical requirements.

Kind regards,

Giuseppe Vittorio De Socio, MD, PhD

Academic Editor

PLOS ONE

Additional Editor Comments (optional):

Reviewers' comments:

Reviewer's Responses to Questions

**Comments to the Author**

1. If the authors have adequately addressed your comments raised in a previous round of review and you feel that this manuscript is now acceptable for publication, you may indicate that here to bypass the “Comments to the Author” section, enter your conflict of interest statement in the “Confidential to Editor” section, and submit your "Accept" recommendation.

Reviewer #1: All comments have been addressed

2. Is the manuscript technically sound, and do the data support the conclusions?

Reviewer #1: Yes

3. Has the statistical analysis been performed appropriately and rigorously? 

Reviewer #1: Yes

4. Have the authors made all data underlying the findings in their manuscript fully available?

Reviewer #1: Yes

5. Is the manuscript presented in an intelligible fashion and written in standard English?

Reviewer #1: Yes

6. Review Comments to the Author

Reviewer #1: (No Response)

7. PLOS authors have the option to publish the peer review history of their article (what does this mean?). If published, this will include your full peer review and any attached files.

Reviewer #1: **Yes: **Giulia Marchetti

---

## [Editor Report · Acceptance letter]

11 Jan 2021

PONE-D-20-19537R1 

Role and effects of zinc supplementation in HIV-infected patients with immunovirological discordance: A randomized, double blind, case control study. 

Dear Dr. Silva:

I'm pleased to inform you that your manuscript has been deemed suitable for publication in PLOS ONE. Congratulations! Your manuscript is now with our production department. 

Kind regards, 

on behalf of

Dr. Giuseppe Vittorio De Socio 

Academic Editor

PLOS ONE